# Roll-your-own cigarette use and smoking cessation behaviour: a cross-sectional population study in England

Sarah E Jackson, Lion Shahab, Robert West, Jamie Brown

Department of Behavioural Science and Health, University College London, London, UK

**Correspondence to**
Dr Sarah E Jackson;
s.e.jackson@ucl.ac.uk

## ABSTRACT

**Objectives** Roll-your-own (RYO) cigarettes have become popular in the UK and reduce the cost of smoking, potentially mitigating the impact of tax increases on quitting. We examined whether RYO cigarette use was associated with reduced motivation to quit smoking, incidence of quit attempts and quit success.

**Design** Cross-sectional survey.

**Setting** England.

**Participants** 38 590 adults who reported currently smoking or having stopped within the past 12 months.

**Main outcome measures** Motivation to quit smoking, quit attempt in the last year, motives for quitting and quit success were regressed onto RYO cigarette use, adjusting for sociodemographic variables and level of cigarette addiction. Mediation by weekly spending on smoking was tested.

**Results** Compared with manufactured cigarette smokers, RYO smokers had lower odds of high motivation to quit (OR=0.77, 95% CI 0.73 to 0.81) or having made a quit attempt (OR=0.87, 95% CI 0.84 to 0.91). Among those who had attempted to quit smoking, quit success did not differ by cigarette type (OR=1.00, 95% CI 0.89 to 1.12), but RYO smokers were less likely to report cost of smoking as a motive to quit (OR=0.68, 95% CI 0.61 to 0.74). Spending on smoking mediated the association between RYO use and quit attempts (β=−0.02, SE=0.003, 95% CI −0.03 to −0.02).

**Conclusions** In England, compared with smokers of manufactured cigarettes, RYO cigarette smokers appear to have lower motivation to quit and lower incidence of quit attempts but similar success of quit attempts. The lower cost of RYO smoking appears to mediate the lower incidence of quit attempts among RYO users.

### Strengths and limitations of this study

► A large, representative sample of the English population.
► Aggregated data from monthly surveys spanning 9.5 years, eliminating potential bias from seasonal differences in the rate of quit attempts.
► The assessment of the most recent quit attempt relied on recall of the last 12 months.
► Quit success was measured by self-reported abstinence, which in randomised trials would be a significant limitation because smokers who receive active treatment may feel social pressure to claim abstinence but is not considered too problematic in population surveys.

## INTRODUCTION

Smoking is one of the leading risk factors for premature death, disease and disability, killing an estimated 6 million people globally each year.[1] Raising taxes to increase the price of cigarettes is consistently reported as one of the most effective ways to prevent smoking uptake, reduce consumption, encourage quit attempts and increase rates of smoking cessation.[2–5] In Europe, a 10% price increase is estimated to result in a 5%–7% reduction in cigarette consumption,[6] with the young and those who are socioeconomically disadvantaged most responsive to changes in price.[5 7]

In the UK, progressive tax increases on tobacco have seen retail cigarette prices rise to among the highest in Europe.[8] However, the potential public health benefits of tobacco tax increases are undermined by the availability of lower cost alternatives such as roll-your-own (RYO; also known as hand-rolled) tobacco. RYO cigarettes are often substantially less expensive than manufactured cigarettes, and a higher proportion is obtained illicitly.[9] The tobacco industry perpetuates this price gap by differentially shifting tax increases between brand segments such that taxes on the cheapest products are not always fully passed onto consumers while taxes on more expensive brands are consistently 'overshifted' with price rises over and above the tax increase.[10] Consequently, while some smokers (particularly those with low disposable incomes) will cut down or quit smoking in response to price increases,[2–5] others may compensate by switching from manufactured to RYO cigarettes to reduce the cost of smoking. Indeed, studies in the UK and other high-income countries have shown that while overall smoking prevalence and manufactured cigarette consumption

are declining, RYO use is increasing, particularly among younger smokers.[11–14]

Despite their rising popularity,[11] the evidence base on RYO cigarettes is relatively scant. A number of studies have described the characteristics of RYO users as younger, male and more socioeconomically disadvantaged,[11 14–16] but we know little about their attitudes towards cessation and quitting behaviour. With RYO cigarettes providing a more affordable option, users may be less inclined to quit. Besides price, RYO cigarettes offer smokers greater control over a variety of aspects, including the weight and diameter of each cigarette, use of a filter and the packing density and amount of tobacco. Indeed, smokers have reported conserving tobacco and rolling thinner cigarettes to reduce the impact of increased excises taxes while presumably also altering smoking behaviour to titrate nicotine intake.[17 18] Evidence suggests that many RYO users perceive RYO cigarettes to be less harmful than manufactured cigarettes[19] when this is not the case,[20] which may reduce their motivation to quit for health reasons.

A few studies have shown that the availability and use of cheap tobacco (including RYO) is associated with lower rates of smoking cessation.[21–23] Data from the International Tobacco Control (ITC) Four-Country Survey indicated that RYO users in the UK and Australia were less likely to report an intention to quit smoking than manufactured cigarette smokers, but there was no association between RYO cigarette use and intention to quit in Canada or the USA.[14] RYO cigarette smokers in the ITC Four-Country Survey were also less likely to make a quit attempt than smokers who did not use RYO or other discounted tobacco, although this difference was not significant after adjustment for sociodemographics and heaviness of smoking.[21] No differences in quit success between manufactured and RYO cigarette smokers were observed in the ITC Four-Country Survey[21] or in another study of ~2000 smokers in New Zealand.[24]

The present study used data from a large population-based sample of English adults to explore whether use of RYO cigarettes is associated with reduced motivation to quit, incidence of quit attempts and quit success, and if so, whether these associations are mediated by spending on smoking.

Specifically, this paper addressed the following questions:

1. How does the prevalence of a quit attempt in the past year in those who smoke RYO cigarettes compare with those smoking manufactured cigarettes, adjusting for a range of sociodemographic factors?
2. Among current smokers, how does the prevalence of high motivation to quit smoking in those who smoke RYO cigarettes compare with those smoking manufactured cigarettes, adjusting for a range of sociodemographic factors?
3. Among past-year smokers who have made at least one quit attempt, how do success rates and quitting motives relating to cost and health in those who smoke RYO

cigarettes compare with those smoking manufactured cigarettes, adjusting for a range of sociodemographic factors and cigarette dependence?
4. Are any differences in these quitting-related outcomes mediated by spending on smoking?

## METHOD
### Design
Data were from the Smoking Toolkit Study, an ongoing research programme designed to provide information about smoking prevalence and behaviour and factors that promote or inhibit smoking cessation at a population level.[25] The study selects a new sample of ~1700 adults aged ≥16 years (of whom ~450 are smokers) each month using a form of random location sampling. Participants complete a face-to-face computer-assisted survey with a trained interviewer. Full details of the study's methods are available elsewhere, and comparisons with national data indicate that key variables such as sociodemographics and smoking prevalence are nationally representative.[25]

### Patient and public involvement
Patients were not involved.

### Study population
For the present study, we used aggregated data from respondents to the survey in the period from November 2008 (the first wave to assess motivation to quit smoking) to March 2018 (the latest wave for which data were available), who smoked cigarettes daily or occasionally at the time of the survey or during the preceding 12 months.

### Measures
#### Measurement of exposure: RYO cigarette use
Participants reported the number of cigarettes they smoked (currently or before quitting, as relevant) on an average day, and how many of these were hand-rolled. For the purpose of the present analyses, we defined RYO cigarette use as at least 50% of total cigarette consumption, a definition used by other studies examining RYO cigarette usage.[14] This definition was favoured over a minimum number per day (eg, at least one RYO cigarette per day) as it allowed inclusion of non-daily smokers, and the 50% threshold allowed for inclusion of individuals who smoked both RYO and manufactured cigarettes.

#### Measurement of outcomes: motivation to quit, quit attempts, quit motives and quit success
Motivation to quit smoking was assessed using the Motivation To Stop Scale,[26] a single-item measure with seven response options representing increasing motivation to quit. We defined high motivation to quit as a response of 6 or 7, reflecting strong intentions to quit within the next 3 months.

Attempting to quit smoking was defined as having made at least one serious quit attempt in the last 12 months. Quit success was defined as self-reported abstinence at

the time of the survey. Further information on the assessment of these outcomes has been published previously.[27]

Those who reported quit attempts were also asked about a wide range of factors that contributed to their most recent quit attempt. We selected for analysis four factors that might plausibly differ according to the type of cigarettes smoked: '*a decision that smoking was too expensive*'; '*seeing a health warning on a cigarette packet*'; '*health problems I had at the time*'; and '*a concern about future health problems*'. Because these items have only been included in the survey since May 2009, data were only available for a subsample of participants (n=12 573).

### Measurement of potential confounders

All potential confounders were selected a priori. Current smokers reported their average weekly spending on cigarettes or tobacco in pounds sterling. Demographic characteristics assessed were age, sex, social grade (ABC1, which includes managerial, professional and intermediate occupations, vs C2DE, which includes small employers and own-account workers, lower supervisory and technical occupations, and semiroutine and routine occupations, never workers and long-term unemployed) and region (government office region grouped into three categories: northern, central and southern England). We included survey year to take account of changes in tobacco control measures that may have impacted RYO and manufactured cigarette smokers differently. We also included nicotine dependence as a potential confounder for some analyses, operationalised as the strength of urges to smoke in the past 24 hours (from 0 '*not at all*' to 5 '*extremely strong*'). This variable has previously been shown to be a better measure of dependence (more closely associated with relapse following a quit attempt) than other measures in this population.[28]

### Statistical analyses

Simple associations between potential confounders and use of RYO cigarettes were examined using analysis of variance (ANOVA) for continuous variables and Pearson's $\chi^2$ for categorical variables.

For our primary analyses, we used logistic regression to examine associations between RYO cigarette use and (1) motivation to quit among current smokers, (2) quit attempts among past-year smokers and (3) quit success among past-year smokers who had attempted to quit. All models were adjusted for age, sex, social grade, region and survey year. We also adjusted for nicotine dependence (strength of urges to smoke) in the model predicting quit success, as previous research in this sample has shown that it reliably predicts this outcome but is not associated with motivation or quit attempts.[29] Results are presented as adjusted odds ratios (ORs) with 95% confidence intervals (CIs). The manufactured cigarette smoker group was the reference category.

In a subsample of smokers who had participated in the survey since May 2009 and had made a serious quit attempt in the last 12 months, we used logistic regression to explore differences between smokers of RYO and manufactured cigarettes in self-reported motives for their most recent quit attempt, adjusting for potential confounders. Results are presented as adjusted ORs with 95% CIs. The manufactured cigarette smoker group was the reference category.

Where RYO cigarette use was significantly associated with an outcome, we tested for mediation by weekly spending on smoking (figure 1). Establishing mediation requires the mediator to be correlated with the exposure (path *a*) and the outcome (path *b*), so we first tested associations between weekly spending on smoking and RYO use and motivation to quit, quit attempts and quit success using ANOVA. Where these associations were significant, we used the *sgmediation* command in STATA, which calculates total (path *c*), direct (path *c'*) and indirect (path *a×b*) effects and tests the significance of the indirect effect using the Sobel test. We used bootstrapping with 5000 sampling replications to estimate the 95% CI and calculated effect ratios reflecting the proportion of the total

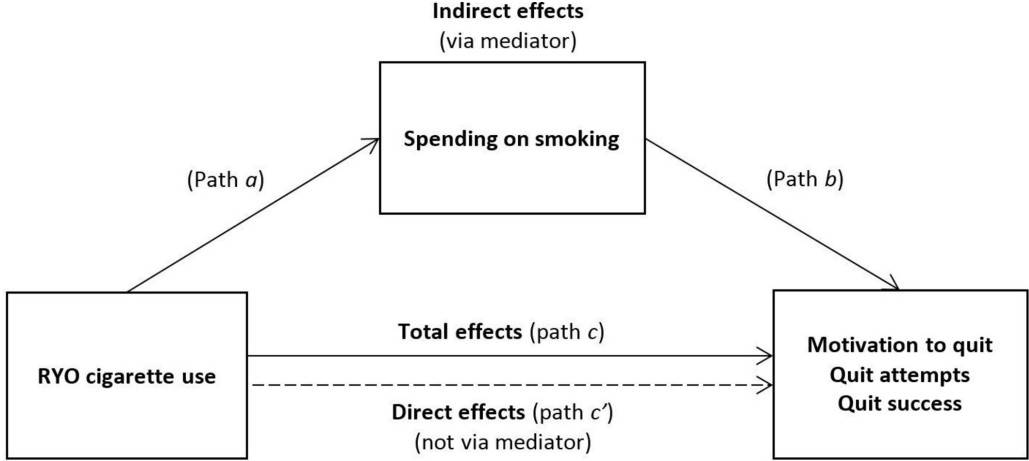

**Figure 1** Mediation model of associations between RYO cigarette use and smoking cessation behaviour via spending on smoking. RYO, roll-your-own.

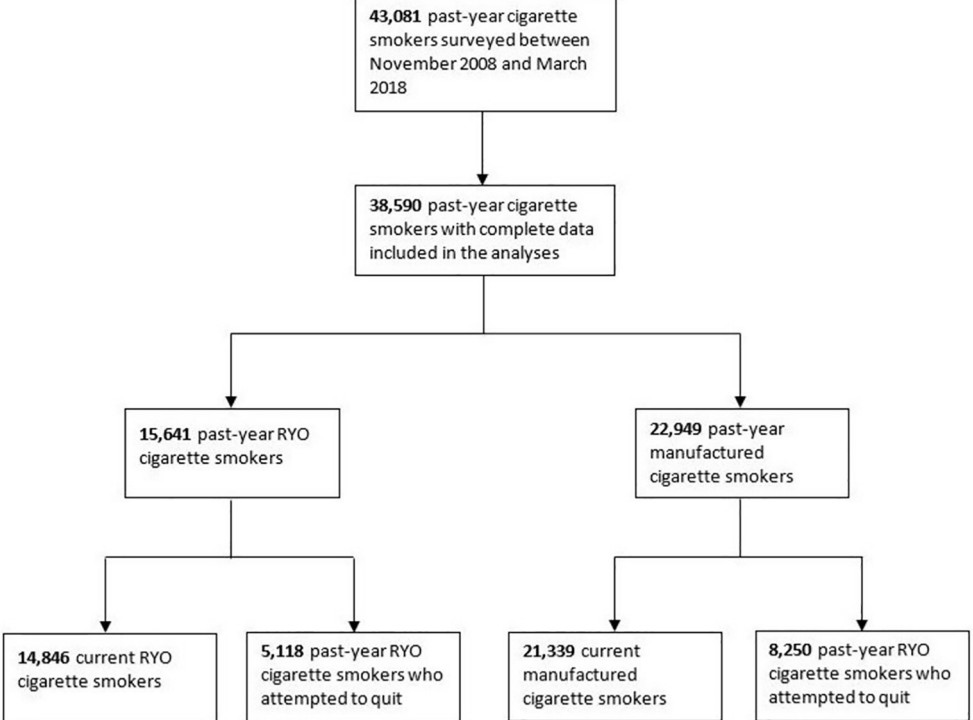

**Figure 2** Summary of sample selection. Note: the groups in the final step are not mutually exclusive but represent overlapping subgroups of the larger groups in the penultimate step. RYO, roll-your-own.

effect of the independent variable on the dependent variable that is explained by the mediator. Mediation models were adjusted for sociodemographics and survey year as previously described and additionally for daily cigarette consumption as we expected this to be strongly related to spending on smoking.

In a sensitivity analysis, we excluded participants who reported smoking both manufactured and RYO cigarettes. In the primary analyses, these individuals were included in the group of RYO cigarette smokers if at least 50% of their cigarettes were hand-rolled, or in the group of manufactured cigarette smokers if less than 50% of cigarettes smoked were hand-rolled.

All analyses were performed using SPSS V.25, with the exception of the mediation models that were run in STATA V.13.

## RESULTS

Figure 2 summarises the selection of the analytic samples. There were 43 081 past-year cigarette smokers surveyed between November 2008 and March 2018. A total of 38 590 (89.6%) provided complete data on cigarette consumption, recent quit attempts and on confounding variables and were included in the present analyses. Of these past-year smokers, 56.3% reported smoking only manufactured cigarettes, 36.6% only RYO cigarettes and 7.1% both manufactured and RYO cigarettes. In the latter group, the percentage of cigarettes that were RYO varied significantly (range 1%–97%) with a mean of 46.2% (SD=22.5). Applying the criterion of at least

50% of cigarettes smoked being RYO resulted in a sample of 15 641 (40.5%) RYO cigarette smokers, with the remaining 22 949 (59.5%) classed as manufactured cigarette smokers. Of past-year smokers, 36 185 (93.8%) were current smokers and provided data on their current motivation to quit. A total of 13 368 (34.6%) had attempted to quit and provided data on the success and motives for the quit attempt.

Sociodemographic and smoking characteristics by RYO cigarette use among past-year, current and those who had attempted to quit in the last year are described in table 1. Of past-year smokers, RYO cigarette smokers were on average slightly younger and a higher proportion were men, from a lower social grade and from southern England than manufactured cigarette smokers (all p<0.001). RYO cigarette smokers reported slightly stronger urges to smoke (p<0.001) and smoked on average one cigarette per day more than manufactured cigarette smokers (13.0 vs 11.9, p<0.001), but their weekly spend was only around half that of the manufactured cigarette smokers (£14.33 vs £26.79, p<0.001). This pattern was similar among current cigarette smokers and those who had attempted to stop in the past year.

The results of the adjusted logistic regression models are shown in table 2. Relative to manufactured cigarette smokers, RYO cigarette smokers were significantly less likely to report high motivation to quit (15.9% vs 20.3%, OR=0.77, 95% CI 0.73 to 0.81) and were less likely to have made a serious quit attempt in the last 12 months (32.7% vs 35.9%, OR=0.87, 95% CI 0.84 to 0.91). However, among

**Table 1** Sample descriptive characteristics: mean±SD or % (n)

| | Past-year smokers (n=38 590) | | | Current smokers (n=36 185) | | | Past-year smokers who attempted to quit (n=13 368) | | |
|---|---|---|---|---|---|---|---|---|---|
| | RYO cigarette smokers (n=15641) | Manufactured cigarette smokers (n=22949) | P values | RYO cigarette smokers (n=14 846) | Manufactured cigarette smokers (n=21 339) | P values | RYO cigarette smokers (n=5118) | Manufactured cigarette smokers (n=8250) | P values |
| Age (years) | 40.72±16.28 | 43.24±17.25 | <0.001 | 40.79±16.29 | 43.30±17.27 | <0.001 | 38.32±15.44 | 40.92±16.24 | <0.001 |
| **Sex** | | | | | | | | | |
| Men | 60.0 (9384) | 45.2 (10376) | <0.001 | 60.0 (8906) | 45.2 (9639) | <0.001 | 56.8 (2909) | 43.6 (3598) | <0.001 |
| Women | 40.0 (6257) | 54.8 (12573) | – | 40.0 (5940) | 54.8 (11 700) | – | 43.2 (2209) | 56.4 (4652) | – |
| **Social grade** | | | | | | | | | |
| ABC1 | 30.2 (4731) | 39.6 (9095) | <0.001 | 29.6 (4391) | 38.8 (8273) | <0.001 | 32.1 (1641) | 41.0 (3381) | <0.001 |
| C2DE | 69.8 (10910) | 60.4 (13854) | – | 70.4 (10455) | 61.2 (13 066) | – | 67.9 (3477) | 59.0 (4869) | – |
| **Region** | | | | | | | | | |
| North | 32.5 (5089) | 35.2 (8086) | <0.001 | 32.5 (4821) | 35.2 (7506) | <0.001 | 31.5 (1611) | 35.7 (2945) | <0.001 |
| Central | 29.8 (4659) | 29.1 (6675) | – | 29.9 (4443) | 29.2 (6236) | – | 31.0 (1585) | 29.3 (2419) | – |
| South | 37.7 (5893) | 35.7 (8188) | – | 37.6 (5582) | 35.6 (7597) | – | 37.6 (1922) | 35.0 (2886) | – |
| Cigarettes per day (n) | 13.03±9.16 | 11.91±8.29 | <0.001 | 12.92±9.02 | 11.81±8.09 | <0.001 | 12.64±9.03 | 11.86±8.39 | <0.001 |
| RYO cigarettes per day (n) | 12.53±8.98 | 0.19±1.06 | <0.001 | 12.44±8.84 | 0.19±1.07 | <0.001 | 12.05±8.80 | 0.21±1.07 | <0.001 |
| Proportion of RYO cigarettes | 0.96±0.12 | 0.01±0.06 | <0.001 | 0.96±0.12 | 0.01±0.06 | <0.001 | 0.96±0.13 | 0.02±0.07 | <0.001 |
| Strength of urges to smoke | 2.03±1.12 | 1.89±1.14 | <0.001 | 2.10±1.08 | 1.98±1.09 | <0.001 | 2.02±1.19 | 1.89±1.18 | <0.001 |
| Weekly spend on cigarettes (£)* | 14.33±10.74 | 26.79±16.99 | <0.001 | 14.33±10.74 | 26.79±16.99 | <0.001 | 14.68±11.13 | 26.64±16.61 | <0.001 |

Social grade: ABC1 includes managerial, professional and intermediate occupations; C2DE includes small employers and own-account workers, lower supervisory and technical occupations and semiroutine and routine occupations, never workers and long-term unemployed.

Strength of urges to smoke: 0 (no urges) to 5 (extremely strong urges).

*In current smokers only.

RYO, roll-your-own.

**Table 2** Multivariable logistic regression models of associations with motivation to quit, quit attempts and (among those who attempted to quit) quit success

| | High motivation to quit | | | Attempted to quit | | | Successfully quit | | |
|---|---|---|---|---|---|---|---|---|---|
| | %* | OR (95% CI) | P values | %† | OR (95% CI) | P values | %‡ | OR (95% CI) | P values |
| Cigarette type | | | | | | | | | |
| Manufactured | 20.3 | 1.00 | – | 35.9 | 1.00 | – | 15.3 | 1.00 | – |
| RYO | 15.9 | 0.77 (0.73 to 0.81) | <0.001 | 32.7 | 0.87 (0.84 to 0.91) | <0.001 | 13.7 | 1.00 (0.89 to 1.12) | 0.988 |
| Age (years) | – | 0.988 (0.987 to 0.990) | <0.001 | – | 0.987 (0.986 to 0.989) | <0.001 | – | 1.010 (1.007 to 1.013) | <0.001 |
| Sex | | | | | | | | | |
| Men | 17.6 | 1.00 | – | 32.9 | 1.00 | – | 15.2 | 1.00 | – |
| Women | 19.5 | 1.10 (1.04 to 1.16) | 0.001 | 36.4 | 1.15 (1.11 to 1.20) | <0.001 | 14.2 | 0.95 (0.85 to 1.06) | 0.344 |
| Social grade | | | | | | | | | |
| ABC1 | 20.2 | 1.00 | – | 36.6 | 1.00 | – | 18.0 | 1.00 | – |
| C2DE | 17.7 | 0.85 (0.80 to 0.89) | <0.001 | 33.7 | 0.89 (0.85 to 0.93) | <0.001 | 12.7 | 0.79 (0.71 to 0.88) | <0.001 |
| Region | | | | | | | | | |
| North | 19.0 | 1.00 | – | 34.6 | 1.00 | – | 15.7 | 1.00 | – |
| Central | 18.3 | 0.95 (0.89 to 1.02) | 0.151 | 35.3 | 1.04 (0.99 to 1.10) | 0.114 | 14.1 | 0.88 (0.77 to 1.00) | 0.052 |
| South | 18.3 | 0.95 (0.89 to 1.01) | 0.119 | 34.1 | 0.99 (0.94 to 1.04) | 0.695 | 14.2 | 0.78 (0.69 to 0.89) | <0.001 |
| Survey year | – | 0.94 (0.93 to 0.95) | <0.001 | – | 0.98 (0.97 to 0.99) | <0.001 | – | 1.02 (0.98 to 1.04) | 0.093 |
| Strength of urges to smoke | – | – | – | – | – | – | – | 0.33 (0.32 to 0.35) | <0.001 |

Social grade: ABC1 includes managerial, professional and intermediate occupations; C2DE includes small employers and own-account workers, lower supervisory and technical occupations and semiroutine and routine occupations, never workers and long-term unemployed.

*Percentage of current smokers in each category who reported really wanting to quit smoking and intending to within the next 3 months.

†Percentage of past-year smokers in each category who had made at least one serious quit attempt in the last 12 months.

‡Percentage of those who had attempted to quit in the last 12 months in each category who were still not smoking after their most recent attempt.

RYO, roll-your-own.

**Table 3** Factors contributing to most recent quit attempt among smokers who had tried to quit in the last 12 months*

| | % of RYO cigarette smokers (n=4891) | % of manufactured cigarette smokers (n=7682) | OR (95% CI)† | P values |
|---|---|---|---|---|
| A decision that smoking was too expensive | 16.0 | 22.0 | 0.68 (0.61 to 0.74) | <0.001 |
| Seeing a health warning on a cigarette packet | 2.9 | 3.2 | 0.84 (0.68 to 1.04) | 0.111 |
| Health problems I had at the time | 18.0 | 13.9 | 1.44 (1.30 to 1.59) | <0.001 |
| A concern about future health problems | 28.3 | 24.9 | 1.15 (1.06 to 1.25) | 0.001 |

ORs reflect the odds of reporting each motive for quitting in the RYO cigarette smoker group relative to the manufactured cigarette smoker group (reference category).
*Subgroup analyses conducted in participants from May 2009 onwards (items not included in previous waves of data collection).
†Adjusted for age, sex, social grade and region.
RYO, roll-your-own.

those who had attempted to quit smoking, there was no significant difference in quit success according to type of cigarettes smoked, with a success rate of 13.7% among RYO cigarette smokers versus 15.3% among manufactured cigarette smokers (OR=1.00, 95% CI 0.89 to 1.12). Younger age, female sex and higher social grade were associated with greater odds of being motivated to, and attempting to, quit. Older age and higher social grade were associated with greater odds of quit success. There was little regional difference in quitting behaviour, with just increased odds of quit success among participants living in northern England. Survey year was negatively associated with odds of high motivation to quit and quit attempts but was not significantly related to quit success. This may reflect the impact of different tobacco control policy changes (eg, banning point of sale displays) at different times across the study period. Nicotine dependence, assessed by self-reported strength of urges to smoke, was strongly associated with reduced odds of quit success.

Analysis of factors contributing to the most recent quit attempt among a subsample of smokers who had tried to quit in the last 12 months is shown in table 3. Data were available for 4891 RYO cigarette smokers and 7682 manufactured cigarette smokers (95.6% and 93.1% of RYO and manufactured cigarette smokers who had made a quit attempt, respectively). RYO cigarette smokers were less likely than manufactured cigarette smokers to report a decision that smoking was too expensive as a motive (16.0% vs 22.0%, OR=0.68, 95% CI 0.61 to 0.74). Few smokers reported health warnings on cigarette packets had motivated their quit attempt, and there was no difference according to the type of cigarettes smoked (2.9% vs 3.2%). Current and future health problems were more frequently cited as contributing factors, but it was RYO cigarette smokers who were more likely to report them as a motive for their most recent quit attempt (current health problems: 18.0% vs 13.9% in manufactured cigarette smokers, OR=1.44, 95% CI 1.30 to 1.59; future health problems: 28.3% vs 24.9%, OR=1.15, 95% CI 1.06 to 1.25).

We explored the possibility that the associations observed between RYO cigarette use and motivation to quit and quit attempts were mediated by spending on smoking (figure 1; as quit success was not associated with RYO use in analysis this was not investigated here). As shown in table 1, RYO cigarette use was associated with significantly lower mean weekly spending on smoking (path *a* in figure 1; p<0.001). Weekly spending on smoking was also positively associated with quit attempts (path *b*; p=0.027) but was not significantly associated with motivation to quit (p=0.533). Thus, mediation analysis was only carried out for quit attempts; results are summarised in table 4 (path *c*, path *c′* and indirect effects in figure 1). There was a significant indirect effect of RYO cigarette use via weekly spend on smoking on the incidence of quit attempts (β=−0.02, SE=0.003, 95% CI −0.03 to −0.02) after adjusting for potential confounders. The effect ratio indicated that weekly spend on smoking explained 100% of the total effect of RYO cigarette use on quit attempts. The direct effect of RYO cigarette use on quit attempts was not significant (β=−0.0003, SE=0.006).

**Table 4** Model testing mediation of the associations between use of RYO cigarettes and quit attempts by weekly spend on smoking (see figure 1)

| | Coeff. | SE | P values* | Bootstrap 95% CI | Effect ratio |
|---|---|---|---|---|---|
| Total effect (path *c*) | −0.02 | 0.006 | <0.001 | – | – |
| Direct effect (path *c′*) | −0.0003 | 0.006 | 0.964 | – | – |
| Indirect effect (via mediator) | −0.02 | 0.003 | <0.001 | (−0.03 to −0.02) | 1.0 |

Analyses are adjusted for age, sex, social grade, region, survey year and daily cigarette consumption.
*P values shown for indirect effects are derived from the Sobel test for consistency with total and direct effects; however, bootstrap 95% CIs provide a more robust indication of significant mediation (see Method for more details).
Coeff, coefficient; RYO, roll-your-own.

Sensitivity analyses that excluded participants who reported smoking both manufactured and RYO cigarettes (n=2750) showed no notable differences in the pattern of results.

## DISCUSSION

The use of RYO cigarettes was associated with reduced motivation to quit smoking and a lower rate of quit attempts but was not significantly related to quit success. RYO users reported spending less each week on smoking than manufactured cigarette smokers and were less likely to cite cost as a trigger for attempting to stop smoking. Spending on smoking was not related to motivation to quit but was a strong mediator of the relationship between RYO use and lower incidence of quit attempts, fully explaining this association.

To our knowledge, no prior studies have examined degree of motivation to quit smoking in relation to RYO cigarette use. However, the ITC Four-Country Survey, which includes nationally representative cohorts of adult smokers from the USA, Canada, the UK and Australia, has assessed intention to quit using a dichotomous yes/no measure. Comparison of RYO and manufactured cigarette smokers revealed lower odds of intending to quit smoking among RYO users in the UK and Australian samples,[14] consistent with our finding that RYO cigarette users were less likely to be highly motivated to quit. However, no significant difference was observed in the USA or Canada,[14] which could be due to cross-national differences in tobacco control policies.

The incidence of quit attempts among RYO and manufactured cigarette smokers in our sample (33% and 36%, respectively) was comparable with results from the ITC Four-Country Survey (34% and 39%, respectively).[21] While in the ITC Four-Country Survey, the adjusted odds of making a quit attempt did not differ significantly according to RYO cigarette use, RYO cigarette smokers in our sample were significantly less likely to report a recent quit attempt even after adjustment for age, sex, social grade and geographic region. This difference may related to difference in the relative cost of RYO and manufactured cigarettes in different jurisdictions.

Rates of quit success were lower in the present sample (14% among RYO smokers and 15% among manufactured cigarette smokers) than in the ITC Four-Country Survey (30% and 31%, respectively)[21] or in a survey of users of Quitline, the largest smoking cessation provider in New Zealand (20% vs 21%).[24] This may be due to differences in the characteristics of smokers between samples. The ITC Four-Country Survey is a longitudinal cohort study with a transparent focus on international tobacco control policy and may attract participants who are more interested in smoking cessation than the average smoker. The New Zealand sample was restricted to smokers who had received telephone counselling for smoking cessation, which is known to improve chances of quit success.[30] By contrast, our sample comprised different monthly random, representative samples of the English population with no requirement for long-term involvement. Despite differences in the prevalence of quit success across studies, results consistently showed no relationship between RYO cigarette use and quit success among those who made a quit attempt.

With previous studies indicating that RYO cigarette smokers often perceive RYO cigarettes to be less harmful than manufactured cigarettes,[19] we had expected health concerns to feature less prominently among RYO cigarette smokers' quit motives as compared with those who smoke manufactured cigarettes. However, results revealed the opposite; RYO cigarette smokers who had attempted to quit were more likely to report current and future health concerns as motives behind their most recent quit attempt. It could be that smokers with greater concerns about the impact of smoking on their health opt to smoke RYO cigarettes as an ineffective attempt to minimise harm.

The cost of smoking was a more prominent motive driving recent quit attempts among smokers of manufactured cigarettes than among RYO cigarette smokers. This is unsurprising given that the average RYO cigarette smoker in our sample smoked one more cigarette each day than the average manufactured cigarette smoker but reported a substantially lower weekly spend on smoking. Importantly, weekly spending on smoking was found to significantly mediate the association between RYO cigarette use and lower incidence of quit attempts, with results indicating that the reason RYO cigarette smokers are less likely to try to quit than those who smoke manufactured cigarettes is the fact that it costs them less. There is a clear need to address the gap in pricing between the most expensive and cheapest cigarettes if policies aiming to reduce smoking via price increases are to achieve their intended effect.

While the cost of smoking was of great importance in determining whether someone was likely to try to quit smoking, weekly spending on smoking was not related to motivation to quit smoking. This is interesting given that cost is the third most cited reason for wanting to quit smoking, after health and social concerns.[31] Moreover, it is seemingly at odds with our finding that spending on smoking was positively associated with quit attempts and with previous studies showing that price increases reduce cigarette consumption.[2–4 6] It is possible that increases in the price of cigarettes have little influence on whether a person *wants* to stop smoking, but rather make it unaffordable to continue to smoke meaning they *need* to stop. The fact that younger and more socioeconomically disadvantaged smokers (groups that tend to have lower disposable incomes) are the most likely to quit in response to price increases[5 7] is consistent with this hypothesis. The distinction between motivation and opportunity as drivers of behaviour is a central component of the COM-B model,[32] a framework that describes how interventions can change behaviour by influencing a person's capability, opportunity and/or motivation. Furthermore, it could be

that cost may have a different impact on routes to quit, primarily increasing unplanned rather than planned quit attempts, which would explain the disconnect with motivation observed here.

A major strength of this study was the use of a large, representative sample of the English population. While previous studies that have examined relations between RYO use and quitting behaviour have had sample sizes of <800 RYO users,[21 24] our sample of >15 000 RYO users provided increased power to detect small effects. In addition, using aggregated data from monthly surveys spanning 9.5 years eliminated potential bias from seasonal differences in the rate of quit attempts.

The study had several limitations. First, we did not have complete data for all past-year smokers surveyed during the study period, and our analysed sample slightly over-represented smokers who were younger, from higher social grades and living in the north of England. Second, the assessment of the most recent quit attempt relied on recall of the last 12 months, introducing scope for bias. Third, quit success was measured by self-reported abstinence. In randomised trials, a lack of biochemical verification would be a significant limitation because smokers who receive active treatment may feel social pressure to claim abstinence. However, in population surveys, the social pressure and associated rate of misreporting is low, and it is considered acceptable to use self-reported data.[33]

In conclusion, smokers who use RYO cigarette are less likely to be motivated to quit or to report having attempted to quit smoking than those who smoke manufactured cigarettes. However, RYO cigarette use appears to be unrelated to quit success among those who do make a serious quit attempt. The cost associated with smoking is a stronger driver of quit attempts among smokers of manufactured cigarettes and mediates the lower incidence of quit attempts among RYO cigarette smokers. While these results provide additional evidence that increasing cigarette prices may encourage people to stop smoking, they also further demonstrate the potential undermining effects of the availability and use of cheap tobacco.

**Twitter** @DrSarahEJackson

**Contributors** All authors designed the study. SEJ wrote the first draft and conducted the analyses. All authors commented on this draft and contributed to the final version. All authors had full access to all of the data in the study and can take responsibility for the integrity of the data and the accuracy of the data analysis.

**Funding** This study was funded by Cancer Research UK (C1417/A22962).

**Disclaimer** The funders were not involved in the study design; in the collection, analysis and interpretation of data; in the writing of the report; or in the decision to submit the paper for publication.

**Competing interests** LS has received honoraria for talks, an unrestricted research grant and travel expenses to attend meetings and workshops from Pfizer and an honorarium to sit on advisory panel from Johnson&Johnson, both pharmaceutical companies that make smoking cessation products, and has acted as paid reviewer for grant awarding bodies and as a paid consultant for healthcare companies. RW undertakes research and consultancy for and receives travel funds and hospitality from manufacturers of smoking cessation medications (Pfizer, GlaxoSmithKline and Johnson&Johnson). JB has received unrestricted research funding from Pfizer. All authors declare no financial links with tobacco companies or e-cigarette manufacturers or their representatives.

**Patient consent** Not required.

**Ethics approval** Approval was granted by the University College London ethics committee.

**Provenance and peer review** Not commissioned; externally peer reviewed.

**Data sharing statement** Extra data are available by emailing Dr Sarah Jackson at s.e.jackson@ucl.ac.uk.

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
