## [Reviewer comments · BMJ Open]

ARTICLE DETAILS

TITLE (PROVISIONAL)	Roll-your-own cigarette use and smoking cessation behaviour: a cross-sectional population study in England
AUTHORS	Jackson, Sarah; Shahab, Lion; West, Robert; Brown, Jamie

VERSION 1 – REVIEW

REVIEWER	Gianluigi Ferrante National Institute of Health, Italy
REVIEW RETURNED	09-Aug-2018

GENERAL COMMENTS	The article is well written, well structured and of good quality. It also highlights a major problem to be considered in tobacco taxation policies, namely that the persistence on the market of cheaper alternatives to manufactured cigarettes diminishes the efficacy of increasing cigarette prices as a means of deterring smoking. I think the article deserves to be considered for publication just for this important message for prevention policies. Some minor revisions are proposed below. a. From the original sample to the analysis sample almost 4500 interviews (about 10%) are lost because of the incompleteness of information on cigarette consumption, recent quit attempts and confounding variables. The 10% is certainly not a worrying percentage, but if it were possible to check that at least for some main variables (sex, age, region of residence, etc.) the incomplete interviews do not differ from the complete ones, you could prove the sample of analysis is not selected. b. A diagram to describe the data of the first paragraph of the Results (page 8, lines 29-45) would be very useful to easily illustrate the information contained in it. c. In the first sentence of the discussion (page 10, lines 23-25) the term 'incidence' is used in an ambiguous way. In fact, the percentages estimated in this survey are not incidences. d. A note defining the two categories of Social Grade is missing in Table 1 and Table 2. The table should be self-explanatory. e. Table 4 lacks a note allowing to correctly interpret the ORs, indicating the reference category
--

REVIEWER	David Young PhD Cancer Council Victoria, Australia
REVIEW RETURNED	28-Aug-2018

GENERAL COMMENTS	I have four concerns that I would like to see addressed: (1) While cost per-se is addressed, titration is not and users of RYO frequently cite their ability to control the amount of tobacco used as an important driver of their choice. I believe that, its effect on cost notwithstanding, titration is worthy of identification as an issue in its own right (2) The statistical analyses sub-section of the Method section is not as clear as it could be. This comment applies to the first two paragraphs of the sub-section and not to the paragraphs dealing with the mediation model. (3) I am not sure I agree with the hypothesis "It could be that smokers with greater concerns about the impact of smoking on their health opt to smoke RYO cigarettes as an attempt to minimise harm until they are ready to quit completely". I think one could equally argue that the reason really equates to 'not cost' (given that it's really a binary choice analysis), or even that the final words "until they are ready to quit completely" could be removed. (4) Survey year is significantly related to "high motivation to quit" and "attempted to quit". This is worthy of comment .
--

VERSION 1 – AUTHOR RESPONSE

Reviewer 1's comments

The article is well written, well structured and of good quality. It also highlights a major problem to be considered in tobacco taxation policies, namely that the persistence on the market of cheaper alternatives to manufactured cigarettes diminishes the efficacy of increasing cigarette prices as a means of deterring smoking. I think the article deserves to be considered for publication just for this important message, useful for prevention policies. Some minor revisions are proposed below.

a. From the original sample to the analysis sample almost 4500 interviews (about 10%) are lost because of the incompleteness of information on cigarette consumption, recent quit attempts and confounding variables. The 10% is certainly not a worrying percentage, but if it were possible to check that at least for some main variables (sex, age, region of residence, etc.) the incomplete interviews do not differ from the complete ones, you could prove the sample of analysis is not selected.

- Response: We have added a line to our limitations section of the discussion mentioning demographic differences between included and excluded participants "...we did not have complete data for all past-year smokers surveyed during the study period, and our analysed sample slightly overrepresented smokers who were younger, from higher social grades and living in the north of England."

b. A diagram to describe the data of the first paragraph of the Results (page 8, lines 29-45) would be very useful to easily illustrate the information contained in it.

- Response: We have added a diagram summarising the sample selection process as suggested (Figure 2) with a footnote.

c. In the first sentence of the discussion (page 10, riche 23-25) the term 'incidence' is used in an ambiguous way. In fact, the percentages estimated in this survey are not incidences.

- Response: We have edited the wording of this sentence to read "The use of RYO cigarettes was associated with reduced motivation to quit smoking and a lower rate of quit attempts, but was not significantly related to quit success."

d. A note defining the two categories of Social Grade is missing in Table 1 and Table 2. The table should be self-explanatory.

- Response: We have added a note defining the social grade categories to both tables.

e. Table 3 lacks a note indicating the reference category, which allows to correctly interpret the ORs.

- Response: We have added a note to this table to aid interpretation of the ORs.

Reviewer 2's comments

I have four concerns that I would like to see addressed:

(1) While cost per-se is addressed, titration is not and users of RYO frequently cite their ability to control the amount of tobacco used as an important driver of their choice. I believe that, its effect on cost notwithstanding, titration is worthy of identification as an issue in its own right

- Response: We now raise the issue of titration in our introduction: "Besides price, RYO cigarettes offer smokers greater control over a variety of aspects, including the weight and diameter of each cigarette, use of a filter, and the packing density and amount of tobacco. Indeed, smokers have reported conserving tobacco and rolling thinner cigarettes to reduce the impact of increased excises taxes, while presumably also altering smoking behaviour to titrate nicotine intake [17,18]."

(2) The statistical analyses sub-section of the Method section is not as clear as it could be. This comment applies to the first two paragraphs of the sub-section and not to the paragraphs dealing with the mediation model.

- Response: We have edited these paragraphs to improve clarity: "For our primary analyses, we used logistic regression to examine associations between RYO cigarette use and (i) motivation to quit among current smokers, (ii) quit attempts among past-year smokers, and (iii) quit success among past-year smokers who had attempted to quit. All models were adjusted for age, sex, social grade, region and survey year. We also adjusted for nicotine dependence (strength of urges to smoke) in the model predicting quit success, as previous research in this sample has shown that it reliably predicts this outcome but is not associated with motivation or quit attempts [26]. Results are presented as adjusted odds ratios (ORs) with 95% confidence intervals (CIs). The manufactured cigarette smoker group was the reference category. In a subsample of smokers who had participated in the survey since May 2009 and had made a serious quit attempt in the last 12 months, we used logistic regression to explore differences between smokers of RYO and manufactured cigarettes in self-reported motives for their most recent quit attempt, adjusting for potential confounders. Results are presented as adjusted ORs with 95% CIs. The manufactured cigarette smoker group was the reference category."

(3) I am not sure I agree with the hypothesis "It could be that smokers with greater concerns about the impact of smoking on their health opt to smoke RYO cigarettes as an attempt to minimise harm until they are ready to quit completely". I think one could equally argue that the reason really equates to 'not cost' (given that it's really a binary choice analysis), or even that the final words "until they are ready to quit completely" could be removed.

- Response: We have edited this sentence removing the words "until they are ready to quit completely", as suggested.

(4) Survey year is significantly related to "high motivation to quit" and "attempted to quit". This is worthy of comment .

- Response: We have added a line to our results highlighting the significant associations with survey year: "Survey year was negatively associated with odds of high motivation to quit and quit attempts, but was not significantly related to quit success. This may reflect the impact of different tobacco control policy changes (e.g. banning point of sale displays) at different times across the study period."

VERSION 2 – REVIEW

REVIEWER	Gianluigi Ferrante National Institute of Health, Italy
REVIEW RETURNED	16-Oct-2018
GENERAL COMMENTS	The main changes proposed at the first review have been implemented. As far as I am concerned, the article can be published.

REVIEWER	David Young Cancer Council Victoria, Australia
REVIEW RETURNED	21-Oct-2018

GENERAL COMMENTS	Much improved. Methodology section significantly improved and all the concerns I voiced appear to have been satisfactorily addressed.
---